# Hyperspectral Imaging Sorting of Refurbishment Plasterboard Waste

**Miguel Castro-Díaz** [1,*], **Mohamed Osmani** [1], **Sergio Cavalaro** [1], **Íñigo Cacho** [2], **Iratxe Uria** [2], **Paul Needham** [3], **Jeremy Thompson** [3], **Bill Parker** [4] **and Tatiana Lovato** [4]

1    School of Architecture, Building and Civil Engineering, Loughborough University, Leicestershire, Loughborough LE11 3TU, UK
2    GAIKER Technology Centre, Basque Research and Technology Alliance (BRTA), Parque Tecnológico de Bizkaia, Edificio 202, 48170 Zamudio, Spain
3    ENVA, Enviro Building, Private Road 4, Colwick Industrial Estate, Nottinghamshire, Nottingham NG4 2JT, UK
4    British Gypsum, East Leake, Leicestershire, Loughborough LE12 6JT, UK
*    Correspondence: m.castro-diaz@lboro.ac.uk

**Abstract:** Post-consumer plasterboard waste sorting is carried out manually by operators, which is time-consuming and costly. In this work, a laboratory-scale hyperspectral imaging (HSI) system was evaluated for automatic refurbishment plasterboard waste sorting. The HSI system was trained to differentiate between plasterboard (gypsum core between two lining papers) and contaminants (e.g., wood, plastics, mortar or ceramics). Segregated plasterboard samples were crushed and sieved to obtain gypsum particles of less than 250 microns, which were characterized through X-ray fluorescence to determine their chemical purity levels. Refurbishment plasterboard waste particles <10 mm in size were not processed with the HSI-based sorting system because the manual processing of these particles at a laboratory scale would have been very time-consuming. Gypsum from refurbishment plasterboard waste particles <10 mm in size contained very small amounts of undesirable chemical impurities for plasterboard manufacturing (chloride, magnesium, sodium, potassium and phosphorus salts), and its chemical purity was similar to that of the gypsum from HSI-sorted plasterboard (96 wt%). The combination of unprocessed refurbishment plasterboard waste <10 mm with HSI-sorted plasterboard ≥10 mm in size led to a plasterboard recovery yield >98 wt%. These findings underpin the potential implementation of an industrial-scale HSI system for plasterboard waste sorting.

**Keywords:** refurbishment plasterboard waste; gypsum recycling; hyperspectral imaging

## 1. Introduction

Plasterboard waste is generated during construction, refurbishment and demolition projects. Refurbishment plasterboard waste usually carries small amounts of other construction materials such as concrete, foam, paint, plastics, wood, ceramics, glass and ferrous metals. It may also contain non-construction materials due to cross-contamination or poor on-site segregation practices. Nowadays, gypsum from refurbishment plasterboard wastes can be recycled through several physical separation methods that remove contaminants. These separation methods comprise manual segregation, grinding, sieving, and ferrous and non-ferrous magnetic separators. Several quality specifications are required for recycled gypsum to be used as feedstock in plasterboard manufacturing. The most important quality parameter of recycled gypsum is its purity in terms of calcium sulphate dihydrate content. The European Union, through the Life+ Gypsum to Gypsum (GtoG) project defined a minimum dihydrate content of 80 wt% for recycled gypsum [1]. This is comparable with the minimum dihydrate content of at least 85 wt% recommended by the Eurogypsum Recycling Working Group [2] and above 85 wt% recommended by the British standard

document BSI PAS 109 [3]. However, recycling of gypsum from refurbishment plasterboard waste for the manufacturing of new plasterboards is still very low in the European Union due to the high degree of contamination. As a result, most of the gypsum from refurbishment plasterboard waste usually ends up in dedicated landfill cells to prevent its decomposition and the release of toxic hydrogen sulphide gas. Recycled gypsum purity depends mainly on the quality of the plasterboard waste received, as current physical recycling processes have limited efficiency and customization possibilities. Furthermore, there are water-soluble chemical impurities present in the gypsum waste, such as chloride, magnesium, sodium, potassium and phosphorus salts, that migrate to the paper-core interface during plasterboard drying and affect paper bonding during plasterboard production [4]. Recently, manual physical segregation combined with a sulfuric acid leaching process has been proven to be an effective technology for producing recycled gypsum from refurbishment plasterboard waste with consistently high chemical purity levels of >96 wt% [5]. However, physical segregation of the contaminants present in the plasterboard waste must be highly effective to obtain recycled gypsum with >96 wt% chemical purity after acid leaching.

Hyperspectral imaging (HSI) is a technique that combines the properties of digital imaging with those of spectroscopy. Algorithms and procedures for spectral data pre-processing, exploration and classification are usually implemented. HSI systems have been used for the identification of materials coming from construction and demolition waste, waste from electric and electronic equipment, municipal solid waste and end-of-life vehicles [6,7]. For instance, HSI systems have been used to separate different types of plastics in municipal solid wastes [8–10]. HSI systems have also been used for the recovery and recycling of concrete, mortar aggregates, bricks, tiles and ceramics present in construction and demolition waste [11], for the separation of concrete, rubber, bricks, wood and plastics in construction waste [12], and to detect gypsum, foam, wood, plastic and brick in demolition waste [13,14]. Other applications include the classification of polyolefin particles in construction waste, and the detection of asbestos in construction and demolition waste [15,16].

The main aim of this work was to determine for the first time whether an HSI-based sorting system can effectively segregate plasterboard mixed with contaminants found in refurbishment plasterboard waste. The specific objectives of this work were to train the HSI-based sorting system with manually segregated refurbishment plasterboard waste prior to testing the system with non-segregated refurbishment plasterboard waste containing contaminants, such as wood, plastics, mortar and ceramics. The impact of HSI-based segregation on gypsum's chemical purity was determined and compared to results obtained with conventional manual segregation. The potential beneficial implications of replacing manual segregation with the automatic HSI-based sorting system would be lower labour costs, higher quality control of the sorted material and higher recycling capacity.

## 2. Materials and Methods

### 2.1. Refurbishment Plasterboard Waste Samples

Two batches of refurbishment plasterboard waste were collected at a skip in Lenton Household Waste and Recycling Centre in Nottingham (United Kingdom). Batch 1 of refurbishment plasterboard waste was collected on 15 October 2020 for HSI system training purposes (Figure 1). Batch 2 of refurbishment plasterboard waste was collected from the same skip on 21 December 2021 and was used to validate the HSI-based classification process. The amount of batch 1 and batch 2 of refurbishment plasterboard waste used in each trial was approximately 10 kg.

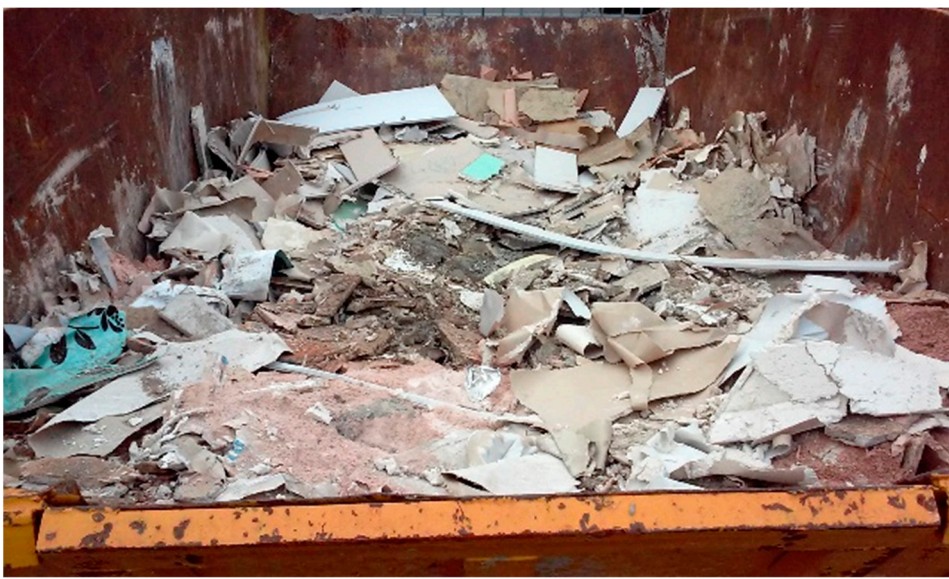

**Figure 1.** Collected refurbishment plasterboard waste in Nottingham (United Kingdom).

*2.2. HSI-Based Sorting System*

Automated in-line sorting of plasterboard contained in refurbishment plasterboard waste was carried out at laboratory scale by means of a table-top setup based on an HSI system. The HSI system included an advanced HSI camera that provides the complete spectral footprint of every pixel of a sample image, air-cooled tubular halogen lamps and a sliding scanning table. The HSI camera (HySpex short-wavelength infrared-384 model, Norsk Elektro Optikk) was the main component of this experimental setup and worked in the spectral range of shortwave infrared (930–2500 nm). The camera is based on the push-broom scanning method for data recording, by which all spectral wavelengths from a narrow line of a spatial scene are simultaneously measured. Table 1 presents the main characteristics of the camera. The illumination system comprised two halogen 100W/12V lamps, one on each side of the camera, oriented at 45°. The sample moving system consisted of a translation stage with a 1040 mm lab rack and controller.

**Table 1.** HSI camera characteristics.

| Description | Specification |
|---|---|
| Spectral range | 930–2500 nm |
| Spatial pixels | 384 |
| Spectral channels | 288 |
| Spectral sampling | 5.45 nm |
| Field of view (FOV) | 16° |
| Pixel FOV across/along | 0.73/0.73 mrad |
| Resolution | 16 bits |
| Maximum speed (at full resolution) | 400 fps |
| Working distance | 21.1 cm |
| Lens | 30 cm close-up lens |

The experimental setup was controlled with a Breeze software package (Prediktera) that allowed the recording of hyperspectral images of materials, training and validation of classification models based on machine learning algorithms, and ultimately, running of sorting applications in real-time.

The principal component analysis (PCA) and the partial least square discriminant analysis (PLS-DA) algorithms are the multivariate data analysis tools that were applied to build the classification models [6,11,16–19]. On one side, PCA is a statistical tool for exploratory data analysis and attempts to find the hidden structure in large and complex

data sets. Hidden structure results from the influence of all variables acting simultaneously and extracting this information reveals, for example, patterns or groupings. Basically, PCA is used to investigate the relationships between samples and measured variables in order to find patterns or groups in the data. It searches for common features, but not for differences between classes. On the other hand, PLS-DA is an algorithm used for the elaboration of linear classification models able to predict the class of unknown samples. The PLS-DA approach consists of building a classification model of all classes, where discrimination is achieved based on inherent class-wise differences. It assumes that any new sample has to belong to just one of the defined classes. For instance, a sample that belongs to class A when the PLS-DA algorithm is applied cannot belong to any other class. The main advantage of PLS-DA is that the relevant sources of data variability are modelled by latent variables, which are a linear combination of the original variables. This allows graphical visualization and understanding of the different data patterns and relations by latent variables scores and loadings [16]. The PLS-DA algorithm for developing classification models of refurbishment plasterboard waste was the only algorithm used in this work because it assigns only one of the available categories, based on its spectral signature, to each unknown sample in the hyperspectral image, making the interpretation of the results easier [17]. Furthermore, this machine learning method has also provided satisfactory results at the laboratory scale with other construction and demolition waste fractions [11,13,14,16].

Predictive classification models based on the PLS-DA algorithm were developed with manually sorted plasterboard fragments 40–50 mm in size (Figure 2a) and other materials typically found in refurbishment plasterboard waste. These other materials included mortar, aggregates, plastics, foam, ceramics, concrete, wood, glass, plastics, metal and rubber. Paper was not classified as a contaminant because plasterboard is constituted by a gypsum core sandwiched between two lining papers. Initially, individual samples were manually positioned on the sliding scanning table and scanned in-line by the HSI camera (Figure 2b).

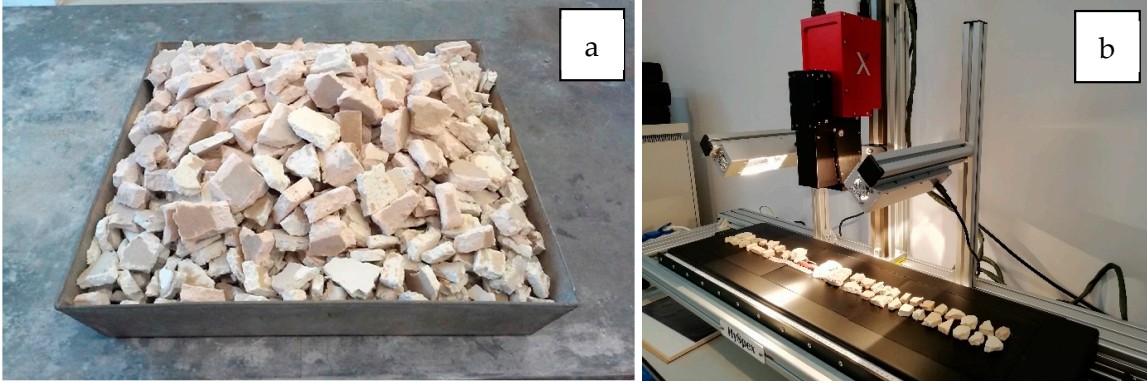

**Figure 2.** Refurbishment plasterboard waste fragments composed of gypsum core and lining paper (**a**) and laboratory-scale experimental setup with advanced HSI camera, air-cooled tubular halogen lamps and sliding scanning table for in-line material classification (**b**).

The methodology for modelling is described in Figure 3 and consisted of recording short-wavelength infrared (SWIR) hyperspectral images of different reference samples by the laboratory setup; adding the class information to the training data set; creating a sample model by means of PCA to remove the background pixels and identify the objects within the images (for instance, plasterboard fragments); and developing the PLS-DA classification model considering the average spectrum of each training sample, the centring pre-treatment of raw spectra and the classes previously defined. The number of measurements or recorded images for model training was 11. The training set consisted of 107 samples made of 51 plasterboard fragments (47.7%) and 56 contaminants (52.3%). The next step was to create a prediction workflow based on the sample and classification models

to analyse and classify unknown materials (test samples—not used for training). According to this workflow, firstly, a measurement (or recorded image) was analysed by the sample model to find or recognize the materials by removing the background. Afterwards, the classification model was applied to the materials in order to predict the class to which the scanned materials belong (plasterboard or contaminants). The model classified individual samples contained in each measurement or recorded image analysing the spectral footprint of each pixel. Thus, the classification approach used was pixel-based.

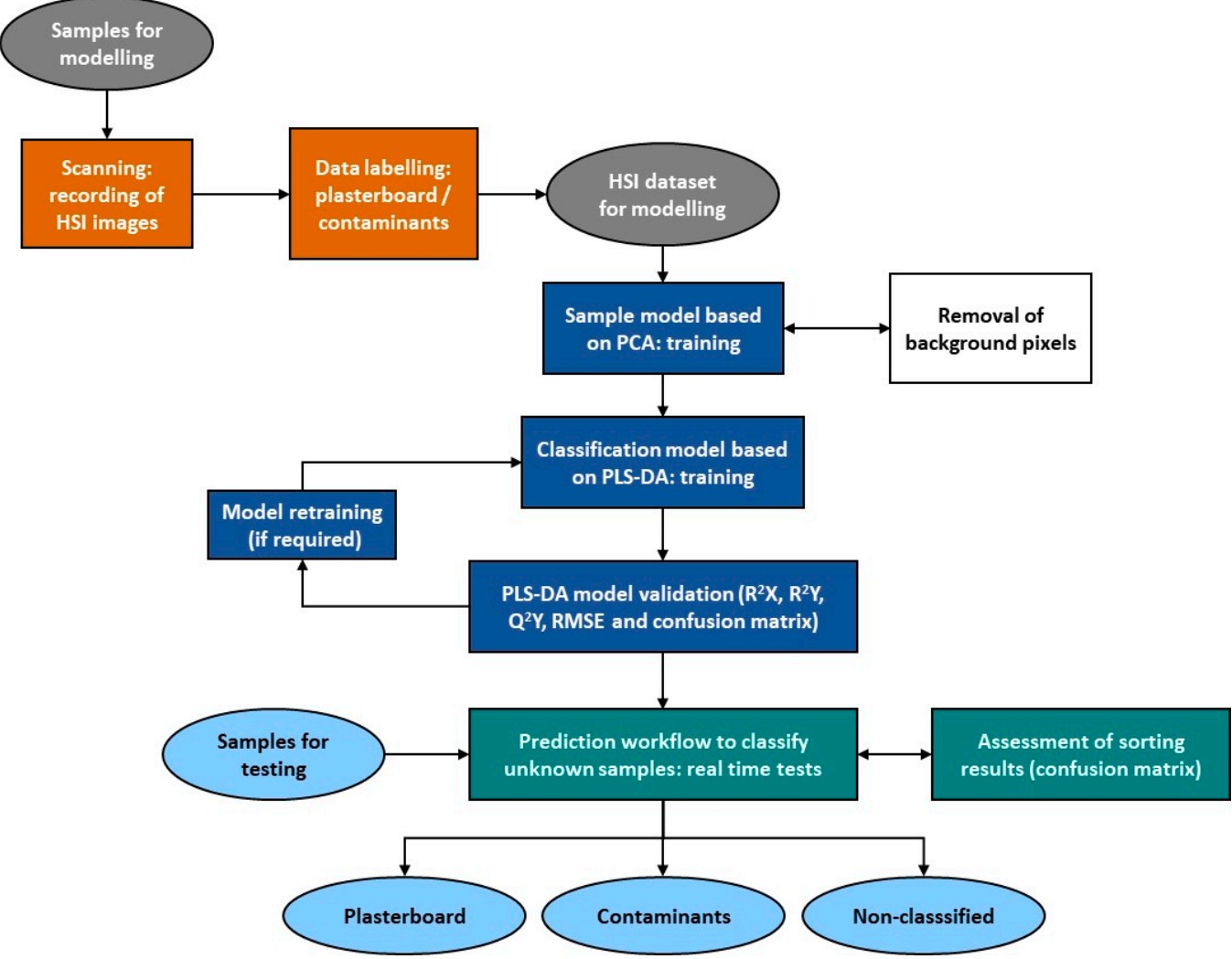

**Figure 3.** Scheme of the HSI methodology.

Additionally, the number of spectral bands used as explanatory variables was reduced to those with the highest contribution to the discrimination. As a result, 111 out of 288 spectral bands were used to train the model, corresponding to the wavelength range 1617.33–2216.68 nm. The spectra of the samples used to train the calibration model by classes are presented in Figure 4.

The correctness or performance of the model was assessed considering the values of $R^2X$ (model fit to the training/known data), $R^2Y$ (model fit to the predicted data), $Q^2Y$ (prediction capacity from cross-validation), RMSE (root mean squared error) and $F_1$ score, and the confusion matrixes for the calibration samples and the laboratory sorting trials. The $F_1$ score was calculated from the precision and recall of the laboratory test using Equations (1)–(3).

$$F_1 \text{ score} = 2 \times (\text{Precision} \times \text{Recall})/(\text{Precision} + \text{Recall}) \tag{1}$$

$$Precision = True\ positives/(True\ positives + False\ positives) \tag{2}$$

$$Recall = True\ positives/(True\ positives + False\ negatives) \tag{3}$$

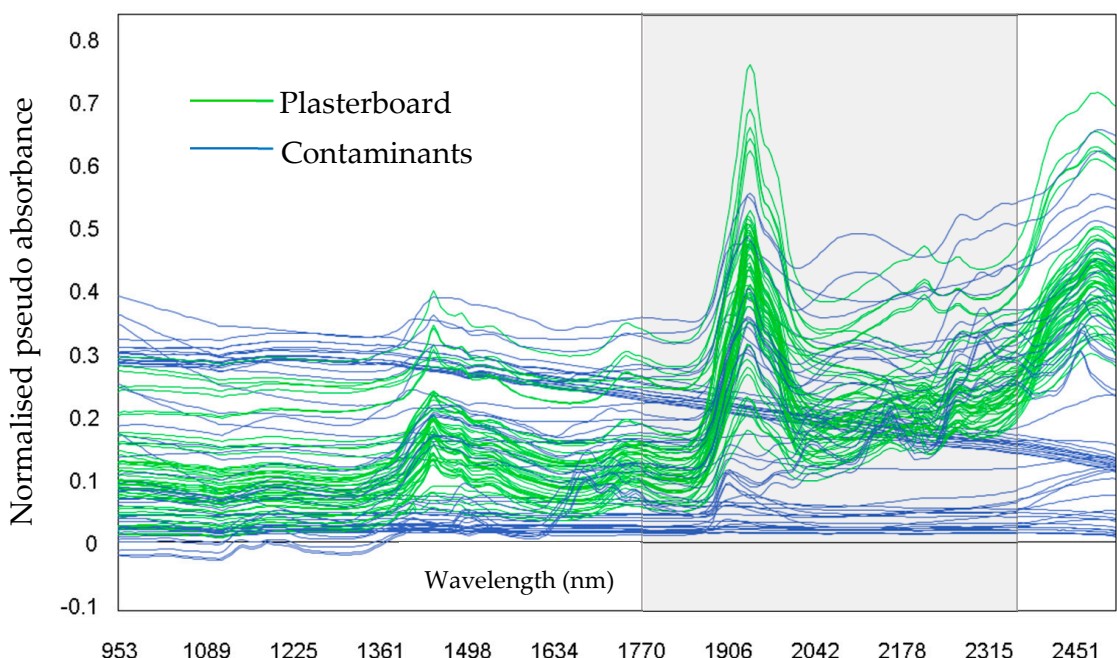

**Figure 4.** Spectra of samples used during model training.

On the other hand, the number of measurements or recorded images for model validation in the real-time classification laboratory test was 113. The validation test set comprised 3766 samples that weighed 8.249 kg, being 91.66 wt% (3536) plasterboard and 8.34 wt% (230) contaminants. An example of a validation trial is shown in Figure 5 and examples of classification images are shown in Figure 6.

| Sample | Group | Image | Real Category | Category | Category |
|---|---|---|---|---|---|
| M0001 (1) | Group | | 🟩 GYPSUM | | 🟩 GYPSUM |
| M0001 (2) | Group | | 🟩 GYPSUM | | 🟩 GYPSUM |
| M0001 (3) | Group | | 🟩 GYPSUM | | 🟩 GYPSUM |
| M0001 (4) | Group | | 🟩 GYPSUM | | 🟩 GYPSUM |
| M0001 (5) | Group | | 🟩 GYPSUM | | 🟩 GYPSUM |
| M0001 (6) | Group | | 🟩 GYPSUM | | 🟩 GYPSUM |
| M0001 (7) | Group | | 🟩 GYPSUM | | 🟩 GYPSUM |
| M0001 (8) | Group | | 🟩 GYPSUM | | 🟩 GYPSUM |
| M0001 (9) | Group | | 🟩 GYPSUM | | 🟩 GYPSUM |
| M0001 (10) | Group | | 🟩 GYPSUM | | 🟩 GYPSUM |

9.8 cm

**Figure 5.** Example of validation trial showing a picture of scanned samples and table of classifications carried out by the PLS-DA model (real category vs. predicted category). Note that plasterboard was defined as 'GYPSUM' in the model software.

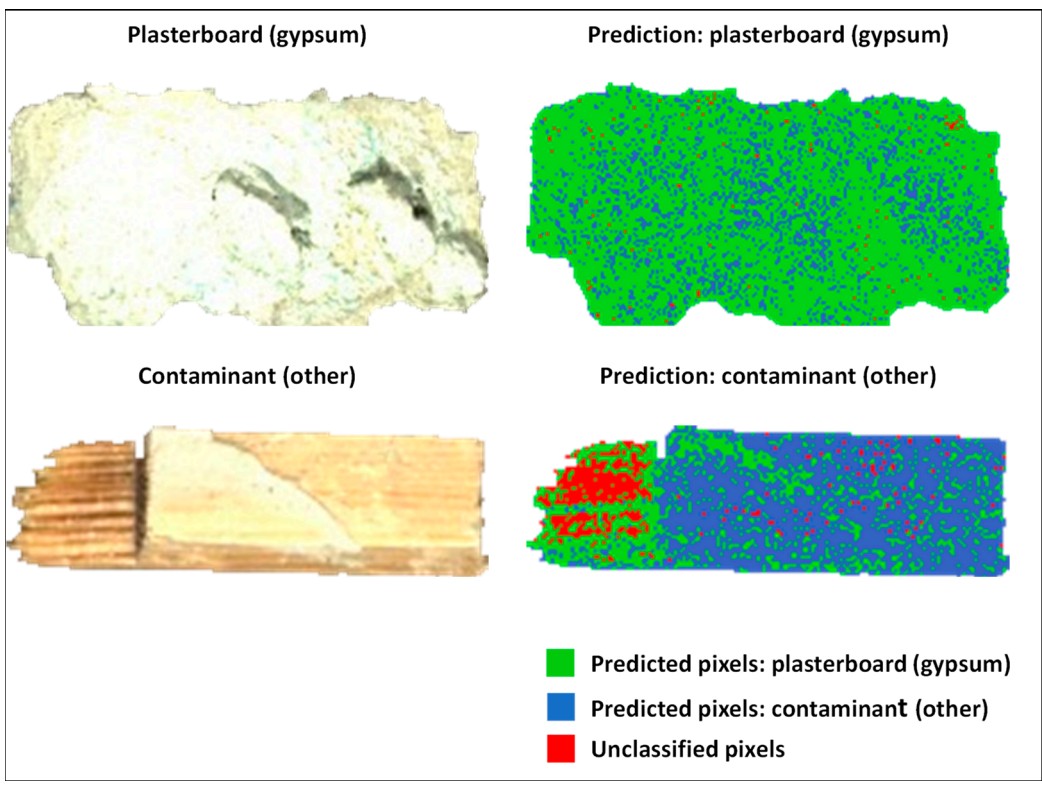

**Figure 6.** Pixel-based prediction maps obtained by applying the PLS-DA model to the refurbishment plasterboard waste in real-time classification laboratory tests (example images).

The input material was classified into three outputs, as outlined in Figure 7. The input material and outputs 1 and 2 were in turn sub-classified into plasterboard and contaminants. Output 3 was recorded to account for small amounts of material that was not classified by the HSI system. In fact, the model software automatically created an additional category (unclassified or no class) to allocate scanned samples that could not be classified as product (plasterboard) or reject (contaminant).

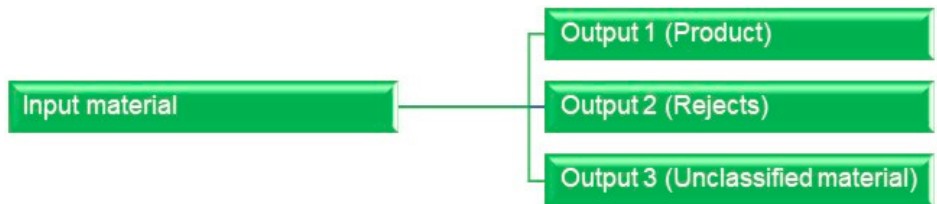

**Figure 7.** Classification of the input material to the HSI-based sorting system into three output streams.

The distribution of the input material among the three output fractions was determined and the components of the product (output 1) and rejects (output 2) were quantified. The overall plasterboard recovery yield was calculated using Equation (4).

$$\text{Recovery yield (wt\%)} = (\text{Plasterboard in Output 1/Plasterboard in Input}) \times 100 \quad (4)$$

### 2.3. X-ray Fluorescence

Manually sorted refurbishment plasterboard waste and refurbishment plasterboard waste sorted with the HSI-based classification process were crushed to obtain gypsum particle sizes <250 microns. This particle size fraction was produced with porcelain mortar and pestle, followed by sieving with a 300 mm sieve according to standards ISO 3310-1 and

BS 410-1. The chemical purity of the samples was determined through X-ray fluorescence (XRF). XRF analyses were performed with an Orbis micro-XRF spectrometer. Sample pellets were prepared by blending 0.8 g of gypsum powder with 0.2 g of boric acid powder (binder). The blend was placed in a die and piston 5 mm in diameter and compacted in a manual hydraulic press applying 10 tons of force to produce the pellet. XRF data were acquired under vacuum in five regions of the pellet using a voltage of 30 kV, current of 0.4 mA, amplifier time of 1.6 µs and acquisition time of 120 s. The weight percentages of $SO_3$, $CaO$, $SiO_2$, $Al_2O_3$, $Fe_2O_3$, $MnO$, $MgO$, $P_2O_5$, $K_2O$, $Na_2O$, $Ni_2O_3$, $SrO$ and $Cl$ were recorded. The chemical purity of the samples was calculated as the sum of $SO_3$, $CaO$, $SiO_2$, $Al_2O_3$ and $Fe_2O_3$ contents expressed as a weight percentage. The mean standard deviation of the chemical purity values was also determined.

## 3. Results

### 3.1. HSI System Training with Manually Sorted Components of Refurbishment Plasterboard Waste

Table 2 shows the composition of the refurbishment plasterboard waste used to train the HSI classification system (batch 1). This composition is not representative of refurbishment plasterboard waste, which can vary greatly depending on the level of contamination. However, this composition provided significant amounts of reference materials (51 plasterboard fragments and 56 contaminants) that enabled the development of the classification model to be implemented in the HSI system.

**Table 2.** Composition of the refurbishment plasterboard waste used for HSI system training.

| Sample | Plasterboard (wt%) | Contaminants (wt%) |
|---|---|---|
| Refurbishment plasterboard waste (batch 1) | 94.81 | 5.19 |

The chemical composition of the gypsum obtained from manually sorted and HSI-sorted plasterboard contained in refurbishment plasterboard waste was determined. The chemical purity of the gypsum obtained from both plasterboard samples was very similar within experimental error (96.65 wt% and 96.85 wt%). These results indicate that the HSI classification system will perform similarly to manual sorting in an ideal case scenario, where plasterboard fragments and contaminants are homogeneously distributed and separated from each other.

### 3.2. Validation of the HSI-Based System for Refurbishment Plasterboard Waste Sorting

Small particles in batch 2 of the refurbishment plasterboard waste were removed with a screen with an aperture of 10 mm. These particles were removed because the laboratory-scale HSI classification process setup was operated manually and the classification of particles <10 mm in size would be a very lengthy process. The sieved material with particles ≥10 mm in size represented 90.73 wt% of the refurbishment plasterboard waste, and thus, small particles <10 mm in size accounted for the remaining 9.27 wt%. Refurbishment plasterboard waste particles ≥10 mm in size were used as input of the HSI-based classification process, whereas particles <10 mm in size were discarded. The input material was placed in the sliding scanning table and analysed by the HSI-based classification system (Figure 2b). The amounts and percentages of the input and output streams and their components were quantified (Table 3). The percentage of paper in the product was 6.1 wt%, which can mostly be removed during subsequent crushing and sieving stages of the physical recycling process. The percentage of contaminants in the product was very small (0.64 wt%). The plasterboard recovery yield of the HSI-based classification system calculated using Equation (4) was 98.53 wt%, which implies that only 1.47 wt% of plasterboard was lost in the rejects. However, 9.27 wt% of the collected refurbishment plasterboard waste had sizes <10 mm, which were discarded as rejects. Therefore, the actual plasterboard recovery yield of the HSI sorting process will depend on whether particles

<10 mm in size present in the refurbishment plasterboard waste are discarded as rejects or not.

**Table 3.** Composition of the input and output materials as determined by the HSI-based classification process.

| Stream | Weight (g) | Weight (%) | Component | Weight (g) | Weight (%) |
|---|---|---|---|---|---|
| Input (≥10 mm fragments) | 8249 | 100 | Plasterboard | 7561 | 91.66 |
| | | | Contaminants | 688 | 8.34 |
| Output 1 (product) | 7498 | 90.90 | Plasterboard | 7450 | 99.36 |
| | | | Contaminants | 48 | 0.64 |
| Output 2 (rejects) | 700 | 8.49 | Plasterboard | 111 | 15.86 |
| | | | Contaminants | 589 | 84.14 |
| Output 3 (unclassified material) | 51 | 0.61 | Not applicable | Not applicable | Not applicable |

Figure 8 shows the contaminants in outputs 1 and 2, the unclassified material found in output 3, and the refurbishment plasterboard waste with particles <10 mm that was not processed. The contaminants in outputs 1 and 2 were mainly wood and mortar. There was also loose paper in these outputs that did not come from the plasterboard. The unclassified material of output 3 was mostly constituted of rubber fragments. The unprocessed material with particles <10 mm in size was mostly constituted of gypsum powder and plasterboard fragments.

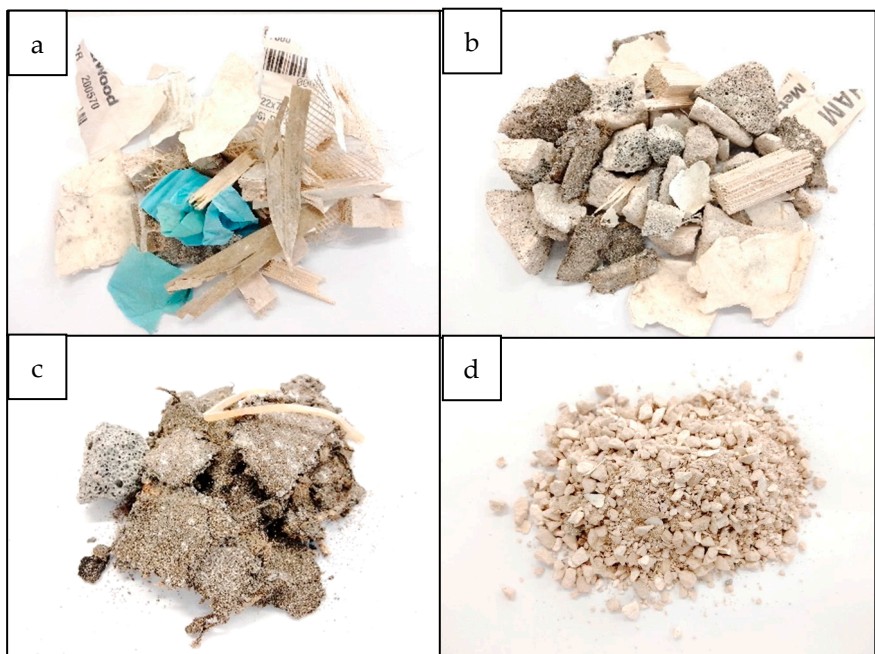

**Figure 8.** Contaminants in output 1 (**a**), contaminants in output 2 (**b**), unclassified material of output 3 (**c**), and unprocessed refurbishment plasterboard waste particles <10 mm in size (**d**).

The chemical purity of gypsum from the discarded refurbishment plasterboard waste fraction <10 mm in size and from the sorted plasterboard in outputs 1 and 2 was determined. Figure 9 shows that the chemical purity of gypsum obtained from unprocessed refurbishment plasterboard waste <10 mm in size and gypsum obtained from the plasterboard in output 1 (product) was about 95.8 wt%. Therefore, the unprocessed refurbishment plasterboard waste could be incorporated into the sorted refurbishment plasterboard waste in output 1 to increase the plasterboard recovery yield without compromising gypsum quality.

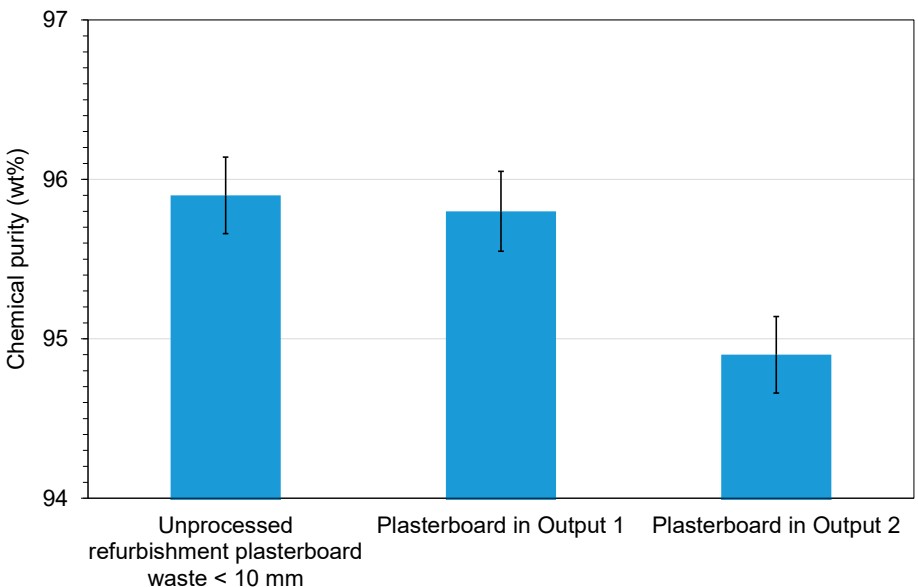

**Figure 9.** Chemical purity of gypsum obtained from refurbishment plasterboard waste fraction <10 mm in size and from HSI-sorted plasterboard in outputs 1 and 2.

By contrast, the chemical purity of gypsum obtained from the plasterboard in output 2 (rejects) is about 1 wt% lower than the chemical purity of the gypsum obtained from the plasterboard in output 1. Overall, the results suggest that HSI-based sorting of refurbishment plasterboard waste and incorporation of the input material <10 mm in size can produce recovery efficiencies above 98 wt% and gypsum's chemical purity close to 96 wt%.

*3.3. Performance Evaluation of the PLS-DA Model in Sorting Refurbishment Plasterboard Waste*

The confusion matrixes for the calibration model and for laboratory-scale refurbishment plasterboard waste sorting are presented in Tables 4 and 5, respectively. These tables show the performance of the supervised learning classification algorithm applied to the calibration samples (training set) and the validation materials (test set), respectively. The confusion matrix allows us to determine if the algorithm is misclassifying materials. It is a comparison between the actual and predicted classification sets. Each row represents the samples in an actual category or class while each column represents the samples in a predicted category. For instance, Table 4 shows that 51 plasterboard training samples (actual category) were predicted as plasterboard by the algorithm (0 as contaminants and 0 as non-classified). Table 5 indicates that the prediction rate was 97.45 wt%.

**Table 4.** Confusion matrix for the calibration model.

| Category | Total Number of Samples (Weight %) | Plasterboard (Weight %) | Contaminants (Weight %) | Non-Classified (Weight %) |
|---|---|---|---|---|
| Plasterboard | 51 (47.7) | 51 (100) | 0 (0) | 0 (0) |
| Contaminants | 56 (52.3) | 0 (0) | 56 (100) | 0 (0) |
| Correct | 107 (100) | 51 (100) | 56 (100) | 0 (0) |

**Table 5.** Confusion matrix for laboratory-scale refurbishment plasterboard waste sorting.

| Category | Total Sample Weight in Grams (Weight %) | Plasterboard Weight in Grams (Weight %) | Contaminants Weight in Grams (Weight %) | Non-Classified Weight in Grams (Weight %) |
|---|---|---|---|---|
| Plasterboard | 7561 (91.66) | 7450 (98.53) | 111 (1.47) | 0 (0.00) |
| Contaminants | 688 (8.34) | 48 (6.98) | 589 (85.61) | 51 (7.41) |
| Correct | 8039 (97.45) | 7450 (98.53) | 589 (85.61) | - |

The cumulative values of the $R^2X$, $R^2Y$ and $Q^2Y$ with ten latent variables and the RMSE value in Table 6, the prediction correctness in Table 7, and the values of the precision, recall and $F_1$ score in Table 8 demonstrate the effectiveness of the PLS-DA classification model.

**Table 6.** Metrics of the PLS-DA classification model.

| Parameter | Value |
|---|---|
| $R^2X$ (cumulative) | 0.99 |
| $R^2Y$ (cumulative) | 0.93 |
| $Q^2Y$ (cumulative) | 0.86 |
| RMSE | 0.13 |

**Table 7.** Samples classification by the PLS-DA model in the laboratory test.

| Classification | Number | Percentage, % |
|---|---|---|
| True positives | 3467 | 92.06 |
| True negatives | 174 | 4.62 |
| False positives | 38 | 1.01 |
| False negatives | 69 | 1.83 |
| No class | 18 | 0.48 |
| Total | 3766 | 100.00 |
| Samples correctly classified | 3641 | 96.68 |
| Samples incorrectly classified | 107 | 2.84 |
| Unclassified samples | 18 | 0.48 |
| Total | 3766 | 100.00 |

**Table 8.** Precision, recall and $F_1$ score of the PLS-DA model in the laboratory test.

| Parameter | Value |
|---|---|
| Precision | 0.99 |
| Recall | 0.98 |
| $F_1$ score | 0.98 |

## 4. Discussion

The HSI classification model shows high effectiveness for its application in refurbishment plasterboard waste sorting. However, there are some potential limitations for the full implementation of the HSI system for plasterboard sorting on an industrial scale. These limitations are listed below, and possible solutions are discussed.

1. The plasterboard and contaminants must have particle sizes ≥10 mm. Plasterboard waste crushing produces significant amounts of gypsum powder and plasterboard fragments <10 mm in size. These particles have been shown to be characterized by high chemical purity (95.9 wt%), which is comparable to the chemical purity of the gypsum obtained from output 1 or product (95.8 wt%). Therefore, particles <10 mm in size should be separated through screening before entering the industrial-scale HSI classification system. Bypassing these <10 mm particles would also minimize the presence of fines in the working environment and safeguard HSI camera performance.

2. The plasterboard and contaminants must be homogeneously distributed on the sliding scanning table because these results strongly depend on the spatial arrangement of the input material. A vibratory screen feeder could be used to separate plasterboard fines and fragments less than 10 mm in size but also to achieve a homogeneous distribution of the input material on the sliding scanning table [20,21]. The design of the vibratory screen feeder components (hopper and vibrating table) and the vibrating table specifications (e.g., angle, vibration frequency and amplitude) should be optimized with the

speed of the sliding scanning table to ensure that there is a homogeneous distribution of the samples for HSI classification. Specifically, the main goal of this optimization process is to achieve a sample monolayer with sufficient distance between different fragments to avoid co-ejection at the end of the HSI classification system and to maximize the HSI system processing capacity. The different shapes and densities of the materials (plasterboard, concrete, wood, ceramics, etc.) must also be considered.

3. Loose paper is not classified as a contaminant because plasterboard is constituted by a gypsum core sandwiched between two lining papers. Lining paper constitutes 4 wt% of the plasterboard and has a density of 0.25–1.52 $g/cm^3$, whereas the gypsum core constitutes around the remaining 96 wt% of the plasterboard and has a density of 2.3 $g/cm^3$ [22,23]. Therefore, the different densities of loose paper and plasterboard fragments should allow for compressed air ejectors to separate these two materials at the end of the HSI system. The lining paper in plasterboard fragments should be removed at the grinding stage of the plasterboard recycling process [24].

The expected economic, health and environmental benefits of using the HSI classification system instead of manual labour for plasterboard waste sorting at refurbishment plasterboard waste recycling sites are listed below:

1. Higher efficiency, reliability and processing capacity;
2. Lower labour costs;
3. Lower handling and exposure to hazardous materials (e.g., sharp objects, dust).

However, there would be additional equipment and energy costs that must be assessed prior to the industrial implementation of this sorting technology.

## 5. Conclusions

A laboratory-scale hyperspectral imaging classification system has been evaluated for the first time as an automatic sorting process for refurbishment plasterboard waste. The HSI classification system was trained with manually segregated components of a batch of refurbishment plasterboard waste. Then, another batch of refurbishment plasterboard waste was processed to validate the HSI classification process. The main conclusions of this study are:

1. The chemical purity of gypsum obtained from HSI-sorted plasterboard and from manually sorted plasterboard was similar;
2. The plasterboard recovery yield attained with the HSI-based sorting system, relative to the plasterboard in the input material, was above 98 wt%;
3. The chemical purity of the gypsum in the refurbishment plasterboard waste particles <10 mm in size was similar to that of the gypsum obtained from the plasterboard in the sorted product (95.8 wt%). Therefore, the unprocessed refurbishment plasterboard waste with particles <10 mm could be combined with the HSI-sorted product material to be used as the feedstock of the sulfuric acid leaching purification process.

Overall, the laboratory-scale HSI-based classification system has been proved to be an efficient sorting process for refurbishment plasterboard waste. The findings reveal the potential industrial application of the HSI-based sorting process for the segregation of refurbishment plasterboard waste, at a lower cost, lower exposure of workers to health risks (e.g., fines) and higher capacity than with manual segregation. Further research should focus on the scalability of the HSI-based sorting process for industrial applications. In this respect, the impact of process parameters, such as conveyor speed, amount of input material on the conveyor and particle size distribution, on plasterboard recovery yield should be determined. Furthermore, potential technical issues that could hinder the efficiency of the HSI-based sorting system, such as the deposition of gypsum dust on the HSI camera over time and operating costs, should be evaluated.

**Author Contributions:** Conceptualization, M.O., S.C., Í.C. and I.U.; methodology, M.C.-D., Í.C. and I.U.; software, Í.C. and I.U.; validation, M.C.-D., Í.C. and I.U.; formal analysis, M.C.-D., Í.C. and I.U.; investigation, M.C.-D., Í.C. and I.U.; resources, Í.C., I.U., P.N., J.T., B.P. and T.L.; data curation, Í.C. and I.U.; writing—original draft preparation, M.C.-D., Í.C. and I.U.; writing—review and editing, M.C.-D., M.O., S.C., Í.C., I.U., P.N. and B.P.; supervision, M.O., S.C. and Í.C.; project administration, M.O., S.C. and Í.C.; funding acquisition, M.O., S.C. and Í.C. All authors have read and agreed to the published version of the manuscript.

**Funding:** This research was funded by the European Union's Horizon 2020 research and innovation program under grant agreement No. 869336.

**Institutional Review Board Statement:** Not applicable.

**Informed Consent Statement:** Not applicable.

**Data Availability Statement:** Not applicable.

**Acknowledgments:** The authors would like to acknowledge the use of XRF facilities within the Loughborough Materials Characterization Centre at Loughborough University.

**Conflicts of Interest:** The authors declare no conflict of interest.

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
