# Peer review of "Hyperspectral Imaging Sorting of Refurbishment Plasterboard Waste"

_applsci, doi:10.3390/app13042413_

Round 1
Reviewer 1 Report
Dear Authors,
I think that you wrote an interesting paper but some clarifications I need.
It is not very clear for me how you appreciate the correctness of the classification model. Did you classify images? How many images? How many sample you used? I think that you should explain the developed procedure with a scheme.
Then I listed below some concerns.
Line
Comment
2. Materials and Methods
I think you should separate Methods from sample preparation paragraph. Moreover, you should better explain what type of equipment you used (brand, model, etc) for HSI analysis
115
What kind of software did you use? Is it a commercial one or you implement your own software?
178
Is this a usual Composition of the refurbishment plasterboard? Why you choose this % for composition of your sample?
222
Can you add size reference?
Author Response
Q1. It is not very clear for me how you appreciate the correctness of the classification model. Did you classify images? How many images? How many sample you used? I think that you should explain the developed procedure with a scheme.I think you should separate Methods from sample preparation paragraph. Moreover, you should better explain what type of equipment you used (brand, model, etc) for HSI analysis.
The correctness or performance of the model was assessed considering, on one hand; the value of R2X, R2Y, Q2Y and RMSE (see Table 6) and, on the other hand, the confusion matrixes for the calibration samples (see Table 4) and the laboratory sorting trials (see Table 5). The model classified individual samples contained in each measurement or recorded image analyzing the spectral footprint of each pixel. The requested information has been added to the manuscript (lines 108-109, 122-145) and the addition of Figures 3 and 5. The 'sample preparation' text has been removed from heading 2.2.
Q2. What kind of software did you use? Is it a commercial one or you implement your own software?
The Breeze software packake was used (line 118).
Q3. Is this a usual Composition of the refurbishment plasterboard? Why you choose this % for composition of your sample?
No. This composition was chosen to have significant amounts of plasterboard and contaminants to validate the HSI system. Table 2 shows the composition of the refurbishment plasterboard waste from which the individual samples of plasterboard and other materials (contaminants) used later for training the system came. Such waste flow provided significant amounts of reference materials (51 plasterboard pieces and 56 impurities) to develop the classification model to be implemented in the HSI system. This has been added in lines 236-240.
Q4. Can you add size reference?
No. Unfortunately, samples and pictures are not available.
Reviewer 2 Report
It's good work. Adding more information on the machine learning algorithm would add a significant advantage to the reader.
More insight into training and testing could greatly help future researchers.
Can the dataset of the experiment be shared with the researchers?
Author Response
Q1. Adding more information on the machine learning algorithm would add a significant advantage to the reader.
This has been added to the manuscript (lines 122-145).
Q2. More insight into training and testing could greatly help future researchers.
This has been corrected with the introduction of lines 159-177 and Figure 3.
Q3. Can the dataset of the experiment be shared with the researchers?
No
Reviewer 3 Report
The manuscript show an interesting topic but the proposed paper are similar to a manuscript entitled "laboratory scale evaluation of hyperspectral imaging sorting of refurbishment plasterboard waste". presented at sum2022 / 6th symposium on circular economy and urban mining / Capri, Italy / 18-20 may 2022. Despite the proceeding was made by the same authors, many pictures and results (Table, figure, consideration ) are the same between this paper and the proceeding.
For this reason, I suggest rejecting the paper
Author Response
Q1. The manuscript show an interesting topic but the proposed paper are similar to a manuscript entitled "laboratory scale evaluation of hyperspectral imaging sorting of refurbishment plasterboard waste". presented at sum2022 / 6th symposium on circular economy and urban mining / Capri, Italy / 18-20 may 2022. Despite the proceeding was made by the same authors, many pictures and results (Table, figure, consideration ) are the same between this paper and the proceeding.
Information in the abstract of applsci-2067553 (1st submission) was incorrect. The information in the conference article was used to produce applsci-2067553 (1st submission) without a thorough revision. As a result, both manuscripts assumed that paper was a contaminant, and for this reason, all calculations in applsci-2067553 (1st submission) were based on this assumption. However, plasterboard is constituted by a gypsum core with two lining papers, and thus, paper cannot be considered a contaminant because the hyperspectral imaging (HSI) system cannot differentiate between loose paper and paper attached to gypsum. Therefore, all calculations were corrected to reflect that paper is NOT a contaminant, and the abstract and the main body of the manuscript were amended to produce applsci-2067553 (2nd submission).
Other differences between the conference paper and applsci-2067553 (2nd submission) are:
- The experimental section presented in the conference article did not provide enough information for the HSI trials to be replicated by other researchers. The experimental section of applsci-2067553 (2nd submission) included information about: i) the HSI system calibration methodology; ii) the number and type of samples used for HSI training and HSI-based sorting trials; iii) HSI camera specifications.
- The performance of the classification was missing in the conference article but included in applsci-2067553 (2nd submission). This was done by presenting the confusion matrix for the HSI system calibration model, the confusion matrix for the laboratory-scale HSI-based sorting trial, and the metrics of the PLS-DA classification model.
- The economic, health and environmental implications of the HSI-based plasterboard waste sorting results were missing in the conference article but were included in applsci-2067553 (2nd submission).
Reviewer 4 Report
Dear authors,
I really like your approach to automate plasterboard waste sorting. The results seem promising, and it could improve the waste recycling.
The authors proposed a measurement setup and showed with two batches of plasterboard waste how Hyperspectral Imaging (HSI) and Partial Least Square - Discriminant Analysis (PLS-DA) can help to automize the sorting.
The overall structure of the work seems reasonable. Still, there are some parts, which are not easy to follow.
- The abstract should be rewritten. Especially, the part with the small components (<= 10 mm) seems too detailed at this point.
- The message of Fig. 7 is not clear. Does this mean the Pasteboard in Output has still a purity of ~95% but is a very small amount?
- The confusion matrices Tab. 4 and Tab. 5 are unclear. Please add additional description text to the caption how to read it.
- What does cumulative values (in line 249) mean? Is it an overall accuracy?
Other open questions:
- How can the model classify data in class 3 (Unclassified) if there are no samples of this class?
- Could moisture of the samples influence the classification, as this is also visible in this spectrum range?
Further remarks:
- Only one machine learning method was tested without showing how this was selected. Normally, I would expect at least two methods which were compared.
Please see the attached PDF for a full list of comments.
Best regards

Author Response
Q1. The abstract should be rewritten. Especially, the part with the small components (<= 10 mm) seems too detailed at this point.
Information in the abstract of applsci-2067553 (1st submission) was incorrect. The information in the conference article was used to produce applsci-2067553 (1st submission) without a thorough revision. As a result, both manuscripts assumed that paper was a contaminant, and for this reason, all calculations in applsci-2067553 (1st submission) were based on this assumption. However, plasterboard is constituted by a gypsum core with two lining papers, and thus, paper cannot be considered a contaminant because the hyperspectral imaging (HSI) system cannot differentiate between loose paper and paper attached to gypsum. Therefore, all calculations were corrected to reflect that paper is NOT a contaminant, and the abstract and the main body of the manuscript were amended to produce applsci-2067553 (2nd submission).
Q2. The message of Fig. 7 is not clear. Does this mean the Plasterboard in Output has still a purity of ~95% but is a very small amount?
This figure relates to chemical purity of gypsum (plasterboard constituent), not level of contaminants present with the plasterboard
Q3. The confusion matrices Tab. 4 and Tab. 5 are unclear. Please add additional description text to the caption how to read it.
Additional text has been added to the manuscript (lines 310-317).
Q4. What does cumulative values (in line 249) mean? Is it an overall accuracy?
The cumulative values are the overall values obtained for said parameters with a specific number of latent variables or components used to develop the model. In this case, the given values of R2X, R2Y and Q2Y are with ten latent variables. This has been added to the manuscript (lines 330-331).
Q5. How can the model classify data in class 3 (Unclassified) if there are no samples of this class?
The software automatically created an additional category (unclassified) to allocate samples that were unknown by the model, and therefore, they were considered not to belong to any of the defined categories. This has been clarified in lines 205-207.
Q6. Could moisture of the samples influence the classification, as this is also visible in this spectrum range?
Moisture content of the samples was not considered in the study.
Q7. Only one machine learning method was tested without showing how this was selected. Normally, I would expect at least two methods which were compared.
The study was only carried out using the PLS-DA algorithm to develop classification models of refurbishment plasterboard waste since this machine learning method had also provided satisfactory results at laboratory scale with other construction and demolition waste fractions (study out of the scope of this paper). Therefore, there are not experimental data to be included in the current paper related to automated sorting of plasterboard waste using classification methods based on other machine learning algorithms. This has been clarified in lines 122-145.
Round 2
Reviewer 3 Report
The manuscript proposes a case study concerning refurbishment plasterboard waste by Hyperspectral imaging sorting. Considering the response to the previous review I suggest the authors add the following information beforepubblications:
Add references in methods regarding PCA and PLSDA.
Describe the main absorptions visible in figure 4. Please include the chemistries that is underlying these spectral ranges. Please add chemistries responsible for the contribution in parenthesis after the spectral range. (example: 1νCH).
Figure 6 how is the "unclassified materials" determined? I can only se that an unknown will end up in one of the "bins". If the principal is not only over and under 0 then please specify how the classification rule are to be understood. If they are over and under then please specify how you can place any materials in the "unclassified materials" category?
Add details about the performance of the PLSDA classification model. More in detail , to evaluate the results, please consider using F1 Score, Precision and Recall. With the F1 score you get information the total performance and distinguish between specific types of errors (false positives and false negatives).
Figure 7 is well described in the text. Maybe it's not necessary.
An example classification image would be useful to understand the quality of the predictions considering that the authors have carried out a "pixel-based" classification.
It is useful to compare the results in figure 9 with the parametric performance obtained by PLSDA in calibration and cross validation.
Reviewer 4 Report
The revised version provides sufficient information for the reader.
Still, more experiments with different algorithmic approaches would be interesting, but are out of the scope of this work.
Round 3
Reviewer 3 Report
The authors have kindly improved the manuscript. In my opinion it is a good case study correctly presented.
Some references are very old. Given the rapid developments of hyperspectral technologies and data processing, I suggest using works no earlier than 2016.
For example, reference 15 seems to have been superseded as a methodological approach by the same authors with the following publication: Bonifazi, G., Capobianco, G., & Serranti, S. (2018). A hierarchical classification approach for recognition of low-density (LDPE) and high-density polyethylene (HDPE) in mixed plastic waste based on short-wave infrared (SWIR) hyperspectral imaging. Spectrochimica Acta Part A: Molecular and Biomolecular Spectroscopy, 198, 115-122.
Author Response
Three newer references (from 2018 and 2019) have been added to the manuscript in lines 121-144 to explain the methodological approach. These references are 17, 18 and 19 in the latest version of the manuscript. References mentioned in the manuscript from this paragraph have been corrected.